# Effect of Water Immersion on Compressive Properties of Coir Fiber Magnesium Phosphate Cement

**DOI:** 10.3390/polym14245339

**Published:** 2022-12-07

**Authors:** Shimin Wang, Shaozhi Song, Mingyu Huang, Zhujian Xie, Liwen Zhang, Wenzhi Zheng

**Affiliations:** Department of Civil Engineering, Guangzhou University, Guangzhou 510006, China

**Keywords:** magnesium phosphate cement, coir fiber, compression performance, water resistance mechanism

## Abstract

Magnesium phosphate cement (MPC) is a new type of inorganic cementitious rapid repair material, but it has poor toughness and is easy to crack. According to our previous research, these problems can be ameliorated by adding natural coir fiber (CF) into MPC. As coir fiber magnesium phosphate cement (CF-MPC) may be used in humid or rainy areas, its water resistance is an important property in consideration. However, at present, little research has focused on this aspect to provide a good theoretical and experimental basis for the practical application of CF-MPC. In this paper, static compression test and solubility test were used to study the mechanical properties and solubility of CF-MPC under water. At the same time, X-ray diffraction (XRD) and scanning electron microscopy (SEM) were used to test the changes of hydration composition and microstructure of the test specimen, so as to understand the deterioration mechanism of CF-MPC in water. The results suggested that, when compared with CF-MPC cured in air, CF-MPC cured in water is more prone to encounter oblique cracks and through cracks in the compression process. Moreover, with the extension of curing time, the compressive strength and elastic modulus of CF-MPC cured in water will continue to decrease, the concentrations of PH, K^+^, and Mg^2+^ in the curing solution will change significantly, resulting in the gradual decrease in the mass ratio of MgO and MgKPO_4_·6H_2_O in CF-MPC matrix, cracks and pores, and looseness in the microstructure.

## 1. Introduction

Magnesium phosphate cement (MPC) is a new type of inorganic cementitious material with excellent mechanical properties. Compared with ordinary Portland cement (OPC), MPC has the advantages of fast setting, strong adhesion, good volume stability, good resistance to heat and high temperature, excellent compatibility with old concrete, high durability, and wide environmental adaptability [1,2,3,4,5,6]. Furthermore, MPC can be hardened and solidified at very low temperatures with almost no shrinkage [7]. Therefore, it is widely used in the fields of pavement repair, bridge cracking, and dam repair [8]. However, MPC is also a brittle material, which has poor toughness and is easy to crack as those typical cementitious materials. Therefore, steel fibers or other artificial fibers—such as polypropylene fiber and polyethylene fiber—are often added to MPC to mitigate these shortcomings of MPC by sharing the stress of MPC matrix and performing its bridging effect. However, the application of these fibers brings out some unavoidable environmental issues such as the high energy consumption in their production and waste treatment, as well as increased engineering costs due to their high price. Therefore, it is valuable to find a fiber with low energy consumption, low cost, and good mechanical properties to replace traditional artificial fibers to improve the properties of MPC.

Coir fiber (CF), as a natural fiber, shows not only superior physical and mechanical properties—such as low density, good ductility, and high tensile strength [9,10,11,12,13]—but also environmental friendly features of low carbon emission and low production cost. Those shining points promise CF a wide application prospect in natural plant fibers’ reinforcing cement-based composites, which has also been proved and demonstrated in our previous studies which found that the addition of CF can improve the toughness and crack resistance of MPC [14,15,16]. Since CF-MPC application may occur in humid or watery environments, it is necessary to clarify its water resistance. Seehra et al. [17] and Sarkar et al. [18] suggested that the MPC will lose its compressive strength after long-term immersion in water, which however is mitigated with the increase in MPC curing time in dry environments. Li et al. [19] and Liu et al. [20] found that a small amount of unreacted phosphate and hydration products in MPC matrix are corroded and hydrolyzed by water, consequently resulting in the decrease in structural compactness and failure with cracking and component leaching. Yu and Yan [21] found that under water curing conditions, an appropriate amount of polypropylene fiber can resist the strength shrinkage of magnesium phosphate concrete in long-term water contact, but when the fiber content is too high, it is prone to fiber “agglomeration”, which cannot be evenly combined with hydration products in the hardened body, finally causing an increase in porosity and decrease in concrete strength. Fang et al. [22] investigated the effect of glass fiber on MPC’s water resistance and reported that the water resistance of MPC can be obviously improved when 3% of glass fiber is added, accompanied with the minimization of MPC compressive strength and flexural strength loss rate after water immersion. Peng et al. [23] studied the water resistance of polyvinyl alcohol fiber reinforced MPC material and found that the addition of fiber increased the water absorption and porosity of mortar samples, but with a negative impact on the strength of samples in water environments. Although extensive studies above are related to the water resistance of MPC or its composite with artificial fibers, few studies focus on the MPC mixing with natural fibers. Additionally, ion exchange between MPC and water still needs to be explored to illustrate the effect of water on the mechanical performance, microstructure, and hydration of MPC. In order to guide the application of CF-MPC in practical engineering, this study was therefore carried out, aiming at the effects of water immersion on the compressive behavior of CF-MPC with different fiber contents. A static compression test was employed to obtain the compressive performances of CF-MPC by respectively curing in both water and air for 7, 28, and 60 days. The effect of water on CF-MPC’s hydration reaction and microstructure was investigated using the microscopic test technique of X-ray powder diffraction (XRD) and scanning electron microscopy (SEM). Meanwhile, to determine the influencing mechanism of water on the hydration reaction and microstructure, solubility tests with curing age corresponding to 7, 28, and 60 days were designed. The concentrations of Mg^2+^ and K^+^ in the curing solution of CF-MPC and the pH of the solution were observed using both an ion detector and pH detector.

## 2. Experimental Programs

### 2.1. Specimens and Raw Materials

This research entails two types of tests: compressive test and solubility test. The specimen size for testing the CF-MPC stress–strain curve in the compressive test is 100 mm × 100 mm × 100 mm (size I), with the curing age of 28 days, and the size of the specimen for testing the compressive strength of CF-MPC is 40 mm × 40 mm × 40 mm (size II), their curing ages being 7, 28, and 60 days respectively. The prepared CF-MPC adopts the proportion of CF volume contents, which are 0%, 1%, 2%, 3%, and 4% respectively. Three specimens with the same parameters are tested in each fiber content group, with two curing methods of immersion curing in water and natural curing in air. The CF-MPC specimens in the solubility test have a size of 40 × 40 × 40 mm (size II), and curing ages of 7, 28, and 60 days respectively. Their fiber contents are in agreement with the mechanical test, with four specimens of the same parameters in each fiber content group. After soaking and curing, the curing solution and specimens are sampled for testing. In the specimen number, A represents air curing, W represents water curing, CF1 represents 1% fiber content, and T7 represents 7 days of curing. More details of the tests and the specimen are listed in Figure 1.

CF-MPC consists of the raw materials such as potassium dihydrogen phosphate (KH_2_PO_4_), magnesium oxide (MgO), borax (Na_2_B_4_O_7_ · 10H_2_O), fly ash (FA), coir fiber (CF), and water. Among them, potassium dihydrogen phosphate and borax are both industrial grade products produced by Nanjing Jianghua chemical glass Co. Ltd. (Nanjing, China) With the mass percentage over 99.5%, they are both white granules at room temperature and easily soluble in water. Magnesium oxide is provided by Henan Xinmi Zhengyang foundry material factory (Zhengzhou, China), which is calcined at 1700 °C to form the dark yellow re-calcined magnesium oxide, with a fineness of 300 mesh, the density of 2.65 g/cm^3^, the specific surface area of 2275 cm^2^/g. Fly ash is supplied by Gongyi Yuanheng water purification material factory, which is 390 m^2^/kg in specific surface area, 300 mesh in fitness, 2.7 g/cm^3^ in density, and black in color. Coir fiber is imported from Sri Lanka and purchased through Jiagaocheng Import and Export Trade Co. Ltd. (Shangrao, China) Figure 2 shows the physical drawing of CF-MPC raw materials. The composition and content proportion of magnesium oxide and fly ash are shown in Table 1 and Table 2 respectively, the mechanical properties of coir fiber were measured by the microcomputer-controlled electronic universal testing machine (SANS-CMT4503) through the tensile test, as shown in Table 3.

The pretreatment of raw materials and the configuration of MPC in this experiment are based on the previous research [14,16], Among them, the quantity ratio of MgO and KH_2_PO_4_ is 5:1, the mass ratio of borax and MgO is 10:1, the mass ratio of MgO and fly ash is 4:1, and the water–binder ratio is 15%. The parameters of MPC basic mix ratio are shown in Table 4. During the test, the coconut pulp fiber is cut to 2 cm and added according to the corresponding volume mixing amount. The mixed MPC slurry was injected into the mold and then demolded after 24 h to obtain the prepared specimen. Finally, air curing and water immersion curing were carried out on specimen according to the experimental design.

### 2.2. Test Setup

The mechanical properties of CF-MPC under pressure are measured by static compression testing. The failure mode, stress–strain curve, compressive strength, and other mechanical properties of CF-MPC during immersion curing in air and water are obtained. The test equipment is a universal testing machine, with loading controlled by displacement. During the formal loading, the displacement rate is set at 2 mm/min. In the measurement of the stress–strain curve, the strain change in the specimen during the test is collected by the strain collector, the load change during the test is recorded by the universal testing machine, and the stress–strain curve of the specimen is obtained by sorting out the data after the test is completed. The compressive strength is tested according to GB/t17671-1999 test method for strength of cement mortar [24], with the size of compression interface measuring 40 × 40 mm. The calculation formula of compressive strength Equation (1) is as follows, and the loss rate of compressive strength is calculated according to Equation (2). In addition, the samples obtained from the failed specimens were used for SEM and XRD tests to understand the microstructure and hydration products of CF-MPC, so as to explain the phenomena observed in the test.
*R* = *F*_max1_/*a*^2^(1)
where *R* is the compressive strength, in MPa, *F*_max1_ is the peak load of the specimen under pressure, and *a* is the side length of the compression surface of the specimen, with the size of 40mm.
*L* = (*f_Vt,A_* − *f_Vt,W_*)/*f_Vt,A_* × 100%(2)
where *f_Vt,W_* is the strength of curing in water when the fiber content of the specimen is *V_f_*, and *f_Vt,A_* is the strength of air curing when the fiber content of the specimen is *V_f_*.

The specimen for solubility test was placed in the curing box and soaked with water for 7, 28, and 60 days respectively. After the relevant curing days, 5 mL of specimen curing solution was extracted from the curing box and put into the centrifugal tube, and the pH value of the solution was measured with a desktop pH meter. After 100 times dilution, the Mg^2+^ and K^+^ concentrations in the solution were measured by Agilent ICP-OES 730 ion concentration meter.

## 3. Compressive Behaviors

### 3.1. Failure Mode

Figure 3a,b provide the failure modes of specimens after water immersion curing and natural air curing for 28 days each. As can be seen from the comparison figures (a) and (b), specimens cured in water mainly show the failure modes with the trend of oblique development and penetration through MPC matrix, and specimens cured in air mainly present a failure of longitudinal development. Under the two curing methods, when CF content was 0%, the substrate fell obliquely off the surface of the specimens cured in water; a similar result occurred with the specimens cured in natural air, but along the vertical crack direction. When CF content was 1%, 3%, and 4%, oblique cracks appeared in the specimens cured in water, but no matrix was pressed off. The specimens cured in natural air failed mainly with vertical cracks, yet without matrix separation. Specimens kept in better integrity under both curing methods at a CF content of 2%.

In general, when the specimen is pressed, the specimen cured in water is more prone to oblique failure and the structural stability is worse. When CF content reaches 2%, the generation of oblique cracks and through cracks under water immersion curing is restrained. Under the two curing methods, the specimen mixed with CF suffered lower damage and kept in higher integrity than the specimen without CF. This is because the interfacial adhesion between CF and MPC significantly restrains the transverse expansion deformation of MPC under compression, thereby restricting the development of cracks during the specimen compression process. The anti-deformation ability of MPC is improved, with no fragmentation when MPC is damaged, but this restraint ability will not increase with CF content. When fiber content reaches a certain amount, agglomeration will appear between fibers, resulting in the decline of restraint deformation ability.

### 3.2. Stress–Strain Relationship

The stress–strain curves of CF-MPC under the two curing methods can be obtained through the static compression test results, as shown in Figure 4, and can be divided into two stages—the rising elastic stage and the falling strain softening stage—according to the influence of fiber content. As Figure 4 suggests, after immersion curing, the peak stress point of the stress–strain curve drops down, the peak strain increases, the falling section of the strain softening stage declines steeper and the strain softening interval becomes shorter. When the fiber content is 0% and 1%, the curve displays no strain softening stage and will drop vertically after reaching the peak point.

Figure 5 shows the stress–strain curves of different CF contents under water immersion curing and air curing respectively. It can be seen from the comparison that when the CF content is 0% and 1%, the early elastic stage of CF-MPC overlaps in most areas during the water immersion curing, and the CF-MPC stress–strain curves of both two curing methods drop linearly after reaching the peak point, displaying the brittle failure of the specimen. When the content of CF is 3% and 4%, most of the elastic stages of the stress–strain curve of CF-MPC under both two curing methods overlap, but for the specimens cured in water, not only does the peak stress reduce and peak strain increase, but the strain softening stage is shortened as well, steepening the decline of the curve in the softening stage more sharply. When the content of CF is 2%, the immersion also leads to the decrease in peak stress, the increase in peak strain, and the sharp decrease in the graph line in the softening stage.

### 3.3. Elastic Modulus

Figure 6 shows the study on the elastic modulus of CF-MPC under two curing methods, which shows that water immersion curing leads to the reduction in the CF-MPC elastic modulus. Compared with CF-MPC of the same content in air, the elastic modulus of the test block cured in water is reduced by 9646.3 MPa at 0% content, 8425.4 MPa at 1%, 7703.1 MPa at 2%, 7313.1 MPa at 3%, and 7005.3 MPa at 4%, respectively. Under the two curing methods, the elastic modulus of CF-MPC will decrease with the increase in CF content, but for the test block naturally cured in air, the elastic modulus of MPC with 4% fiber content (15,047 MPa) is 5199 MPa lower than that of MPC with 0% fiber content (20,246 MPa). When cured in water, the elastic modulus of CF-MPC with 4% content is 2558 MPa lower than that of MPC with 0% content.

### 3.4. Compressive Strength

Figure 7 shows the loss rate of compressive strength and strength after immersion of CF-MPC under two curing environments. The compressive strength of CF-MPC cured in water decreases with the increase in immersion time, opposite to the change trend of CF-MPC compressive strength cured in air. When cured in water for 7 days, the strength loss rate of CF-MPC under each fiber content is 0.03% compared with the strength of the specimen cured in air. At this time, the strength reduction in the specimen is not serious, indicating that the effect of short-time immersion on the strength of MPC matrix is low. After 28 days of soaking, the specimens further decrease the compressive strength with a gradually increasing gap, compared with the specimens cured for the same time in air. When soaked for 60 days, the strength of CF-MPC further reduces its strength compared with that of curing for 28 days, while the difference of strength between curing in air for 60 days versus 28 days is small. However, the compressive strengths under the two curing methods still share some commonality in their change; that is, the compressive strength decreases slowly with the increase in CF content, this is because CF is mainly composed of cellulose, lignin, pectin, and hemicellulose. The microstructure is composed of hollow tube bundle fiber. Its structure is not as dense as cement-based materials, although the axial tensile performance is good, but the stiffness is poor and the cement-based physical filling effect is poor. In addition, with the increase in CF content, the volume ratio of MgKPO_4_·6H_2_O (the hydration product of MPC) and MgO (functioned as the framework of MPC in the specimen) decreases. Therefore, the greater the CF content, the faster the MPC strength decreases.

## 4. Microcosmic

### 4.1. Solubility Analysis

Figure 8 shows the changes of K^+^, Mg^2+^, and PH in the solution when CF-MPC is immersed in water. The K^+^ concentration will increase sharply in the early stage and slowdown in the later stage. This is because the unreacted KH_2_PO_4_ and a small amount of K_2_HPO_4_ produced in the preparation process will dissolve a large amount of K^+^ in the early stage. When K_2_HPO_4_ is completely dissolved, the growth trend of K+ concentration will slow down again. The higher the fiber content is, the higher the K^+^ concentration is. This is because when the fiber content is too high, it will increase the matrix pores and make the specimen more affected by water immersion. The Mg^2+^ concentration first increases and then decreases, and finally remains at about 9 mg/L. in the early stage, the H ^+^ formed by KH_2_PO_4_ dissolution and the water immersed in the matrix react with MgO and MgKPO_4_·6H_2_O, resulting in the increase in Mg^2+^ concentration. When the ions involved in the hydration reaction meet on the surface of the matrix or inside the matrix or in the solution, they will react to form MgKPO_4_·6H_2_O again, resulting in the decrease in Mg^2+^ concentration. At the same time, due to the use of still water for water maintenance, the amount of water is limited. When the dissolved amount of MgO reaches the maximum, the Mg^2+^ concentration will not change greatly. In the early stage, the PH of the solution increases with the increase in the curing time of the specimen in water. This is due to the continuous dissolution of MgO and HPO_4_^2−^ in the specimen. The PH growth slows down at 7–28 days. Combined with the decrease in Mg^2+^ concentration, it can be understood that this is due to the consumption of part of OH^-^ in the formation of hydration product MgKPO_4_·6H_2_O. In addition, when the fiber content increases, the microstructure of the specimen becomes worse, which will lead to easier dissolution of various substances from the matrix into water. Therefore, the higher the fiber content in the later stage, the greater the PH.

The two ion concentration change diagrams and PH change diagrams show that the substances in CF-MPC dissolve into water, migrate, and are lost. Finally, in the interior, surface, or solution of the test piece, some substances recrystallize to form MgKPO_4_·6H_2_O. The process of substance migration and loss makes pores form in the MPC matrix, resulting in increased porosity and reduced structural compactness, as shown in Figure 9, resulting in microstructure defects. Thus, its water resistance is poor.

### 4.2. Hydration Analysis

The XRD patterns of CF-MPC cured in air and in water are shown in Figure 10. In analyzing the main components of CF-MPC—i.e., MgKPO_4_·6H_2_O and MgO—it is found that the diffraction peak angles of MgKPO_4_·6H_2_O are mainly in the range of 15°~22° and 27°~35°, and the diffraction peak angles of MgO are mainly in the range of 37°, 43°, 63°, 75°, and 80°, both showing roughly the same angle under two curing methods.

Quantitative analysis was carried out by calculating the total area by integrating the characteristic peaks of XRD, and the obtained ratio of the characteristic peak area of MgO and MgKPO_4_·6H_2_O was the mass ratio of MgO and MgKPO_4_·6H_2_O, as shown in Figure 11. as the strength of MPC is mainly determined by the ratio of MgO to MgKPO_4_·6H_2_O, which is close to 0.6 when the CF-MPC in this test is cured in the air to the ideal hydration degree. It can be seen from the figure that the mass ratio of MgO/MgKPO_4_·6H_2_O in the CF-MPC cured in the air gradually decreases with the increase in curing time, indicating that the underreacted substances in the test block are still undergoing slow hydration reaction, resulting in the increased density in CF-MPC microstructure as in Figure 12 and the enhancement of the CF-MPC compressive strength with the prolongation of curing days. Meanwhile, it can be seen that the ratio of MgO/MgKPO_4_·6H_2_O increases with the addition of fiber content in air curing. This is because the water absorption of coir fiber affects the hydration reaction by reducing the water participating in hydration reaction. In addition, the mass ratio of MgO/MgKPO_4_·6H_2_O is close to—but never reaches—0.6, indicating the existence of underreacted substances in the matrix. In water immersion curing, the mass ratio of MgO/MgKPO_4_·6H_2_O decreases with the increase in immersion curing time. This is because the immersing water enters the micro cracks of the specimen, dissolving soluble substances and further enhancing the reaction of underreacted substances to form MgKPO_4_·6H_2_O, resulting in the still increase in the total amount of MgKPO_4_·6H_2_O in the matrix due to the low solubility of MgKPO_4_·6H_2_O. On the other hand, MgO decreases due to the reaction as in Figure 9a, which reflects the changes of K^+^, Mg^2+^, and PH in Section 4.1. Additionally, the mass ratio of MgO/MgKPO_4_·6H_2_O decreases with the increase in fiber content. In combination with what is described in Figure 12, it can be found that this is because the increase in fiber content reduces the compactness of CF-MPC structure, resulting in easier water immersion in MPC matrix.

### 4.3. Microstructure

Figure 12 is the microstructure diagram of CF-MPC with different CF contents under different curing environment and curing time. It can be seen from Figure 12a that in the MPC with 0% fiber content after 7 days of curing, the MPC structure cured in water is relatively dense, yet still with some holes, while the MPC structure cured in air has poor compactness but fewer holes. According to Figure 11, it is known that this is because of the generated holes in the MPC matrix by the dissolution of some substances in the MPC cured in water for 7 days. In addition, the XRD results in Figure 11 also demonstrate that the underreacted substances are further hydrated under the influence of water immersion, resulting in better structural hydration and enhanced compactness. As for MPC cured in air, although there are few holes in it, its degree of hydration is poor. Therefore, there is little difference in the compressive strength of the specimens under the two curing methods. Additionally, the increase in fiber content assists the occurrence of fiber agglomeration in the specimen, resulting in the appearance of holes and the deterioration of structural compactness.

It can also be seen from Figure 12 that with the increase in curing days, the MPC structure cured in air becomes dense, verifying the decrease in MgO/MgKPO_4_·6H_2_O in Figure 11 and reflecting the further development of hydration reaction. The MPC structure cured in water becomes loose, and more structural holes occur with the increase in fiber content, illustrating the dissolution of soluble substances in Section 4.1. When immersed in water for 60 days, the fibers in CF-MPC with 4% content display the damage of cracking. This is because the pH of the immersed solution increases in alkalinity with the dissolution of the internal substances of MPC (Figure 9). CF-MPC with high fiber content produces more pores, and the internal fibers are corroded after being immersed in alkaline solution for a long time, causing the loss of pectin and other components in the fiber, and resulting in problems such as cracking and damage.

## 5. Conclusions

In this study, the static compression test was used to study the compressive properties of coir fiber magnesium phosphate cement under immersion, and the effects of immersion curing on the failure mode, stress–strain relationship, elastic modulus, compressive toughness index, and compressive strength of magnesium phosphate cement with different coir fiber contents. In addition, the ion concentration of the curing solution was analyzed, and the CF-MPC was sampled for XRD and SEM to detect the hydration degree and microstructure of CF-MPC. Based on the results, the following conclusions can be drawn:

(1)Water immersion will change the failure mode of CF-MPC, resulting in oblique development and cracks penetrating the matrix.(2)When CF-MPC is cured in water, the elastic modulus of the specimen will be greatly reduced, the peak stress will be reduced, and the peak strain will be increased.(3)During water curing, the K^+^ and Mg^2+^ concentration change diagram and PH change diagram portray the dissolution, migration, and recrystallization process of CF-MPC under water immersion curing, and they explain the change in MgO/MgKPO_4_·6H_2_O mass ratio and microstructure defects caused by water immersion.(4)XRD results show that the mass ratio of MgO to MgKPO_4_·6H_2_O in soaked CF-MPC will decrease with the increase in curing time, and the mass ratio of MgO to MgKPO_4_·6H_2_O will decrease with the increase in fiber content at the same soaking time.(5)The results of solubility testing and SEM show that the soaking environment leads to the dissolution and leaching of unreacted phosphate in water environments, which generates additional defects in CF-MPC dense structure. However, excessive fiber content will also lead to the increase in structural holes, and the fibers embedded in MPC will be damaged with the increase in immersion time and the corrosion of alkaline solution.

## Figures and Tables

**Figure 1 polymers-14-05339-f001:**
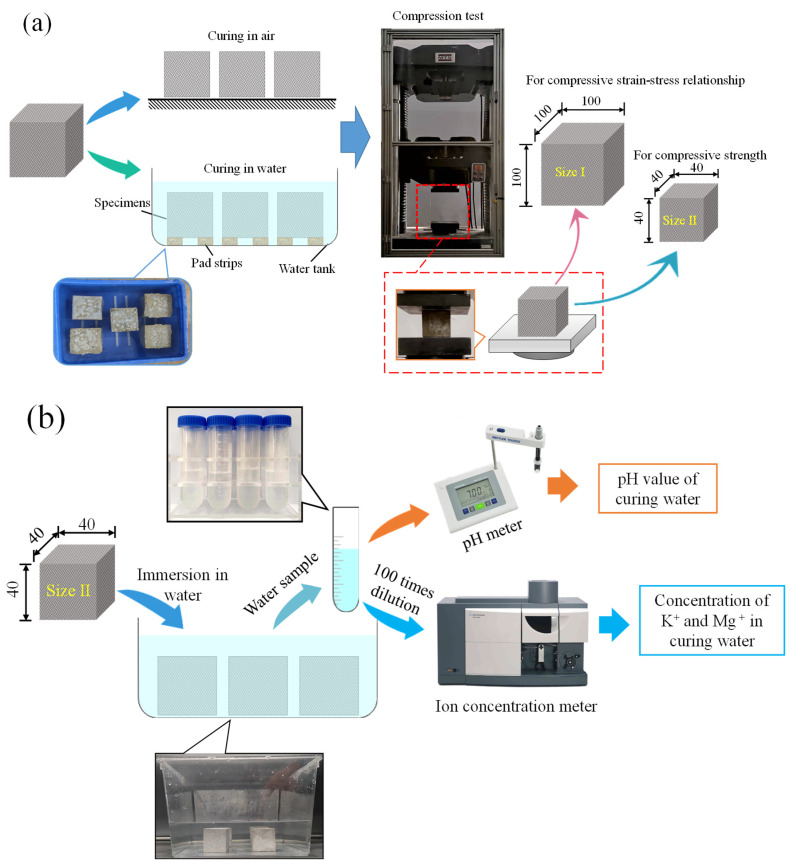
Specimen and test setup (Unit: mm). (**a**) Compressive test; (**b**) Ion concentration test.

**Figure 2 polymers-14-05339-f002:**
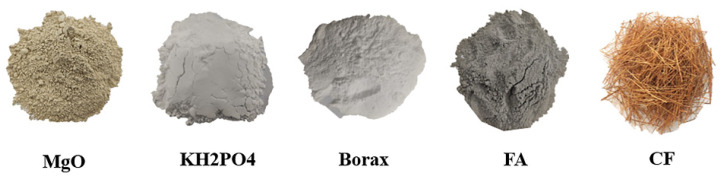
Physical drawing of raw materials.

**Figure 3 polymers-14-05339-f003:**
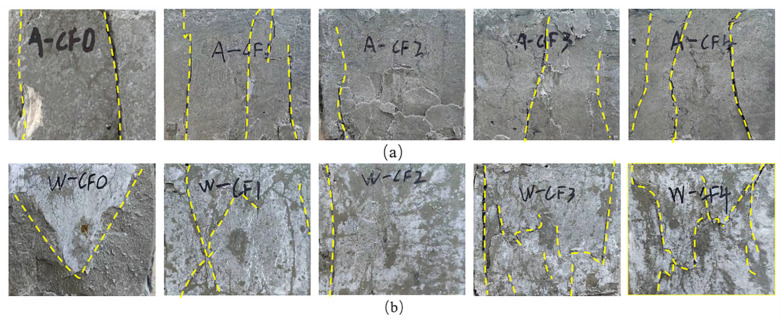
Failure modes. (**a**) A-T28; (**b**) W-T28.

**Figure 4 polymers-14-05339-f004:**
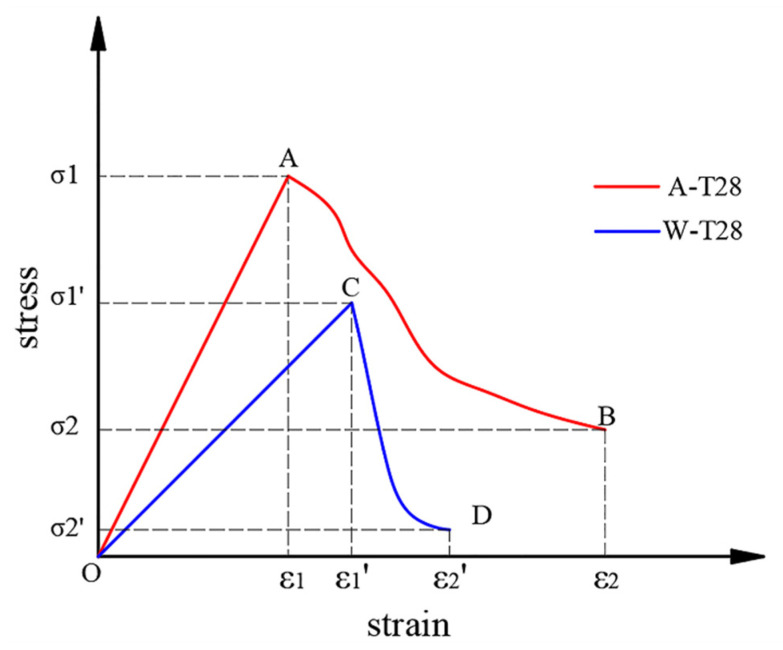
CF-MPC stress–strain curve.

**Figure 5 polymers-14-05339-f005:**
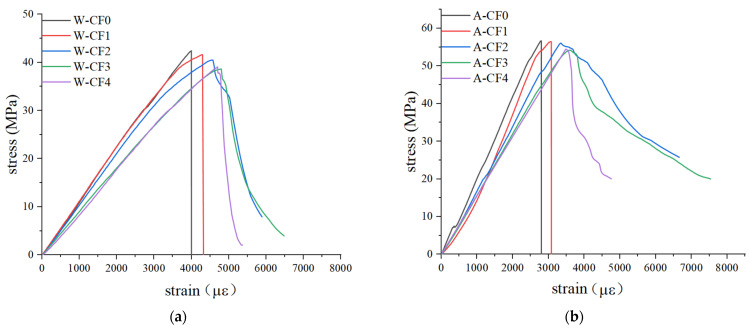
Stress–strain curves of CF-MPC under different curing environments. (**a**) Water immersion curing; (**b**) Natural conservation in air.

**Figure 6 polymers-14-05339-f006:**
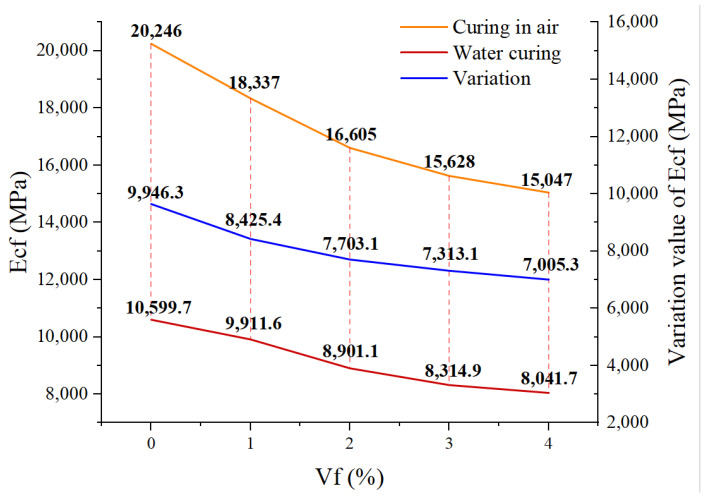
Elastic modulus of CF-MPC.

**Figure 7 polymers-14-05339-f007:**
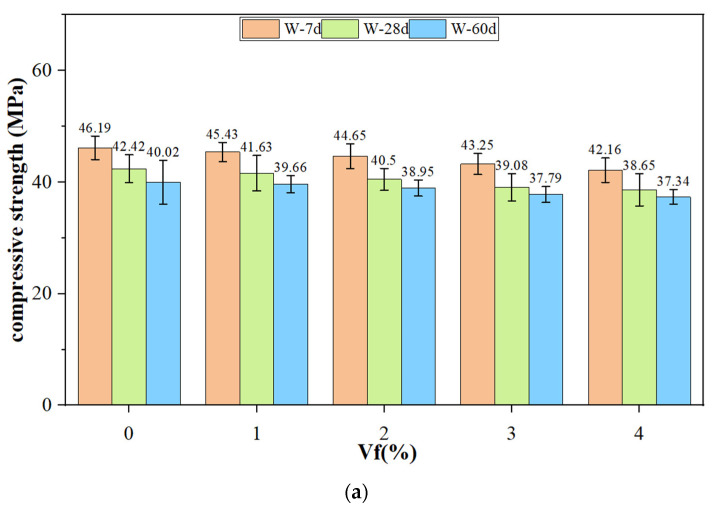
CF-MPC compressive strength and strength loss rate. (**a**) Water immersion curing; (**b**) Natural conservation in air; (**c**) Strength loss rate.

**Figure 8 polymers-14-05339-f008:**
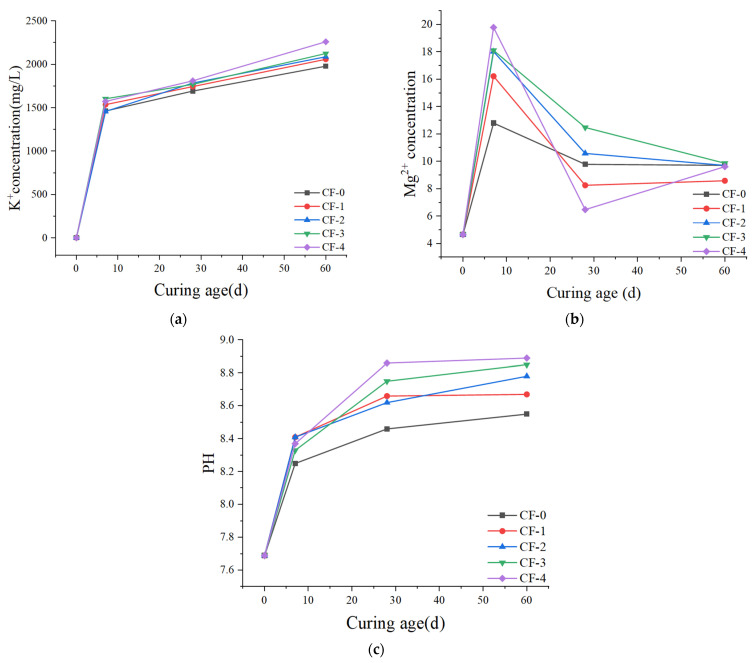
(**a**) K^+^ Concentration variation diagram; (**b**) Mg^2+^ Concentration variation diagram; (**c**) PH change diagram.

**Figure 9 polymers-14-05339-f009:**
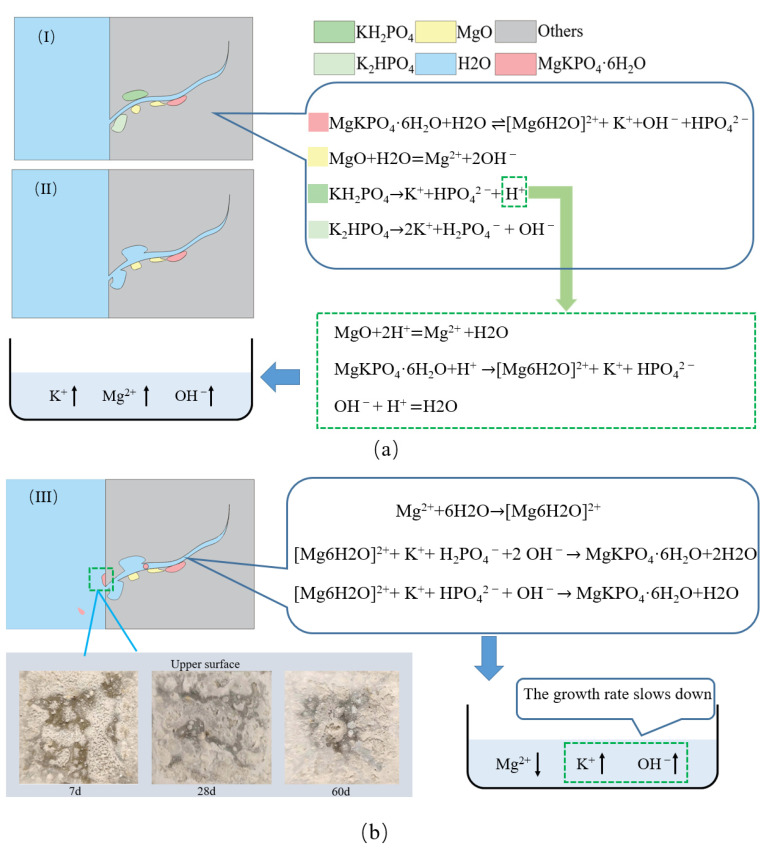
Water resistance mechanism. (**a**) Dissolution of substances (I) and migration of ions (II); (**b**) Recrystallized to form hydration products (III).

**Figure 10 polymers-14-05339-f010:**
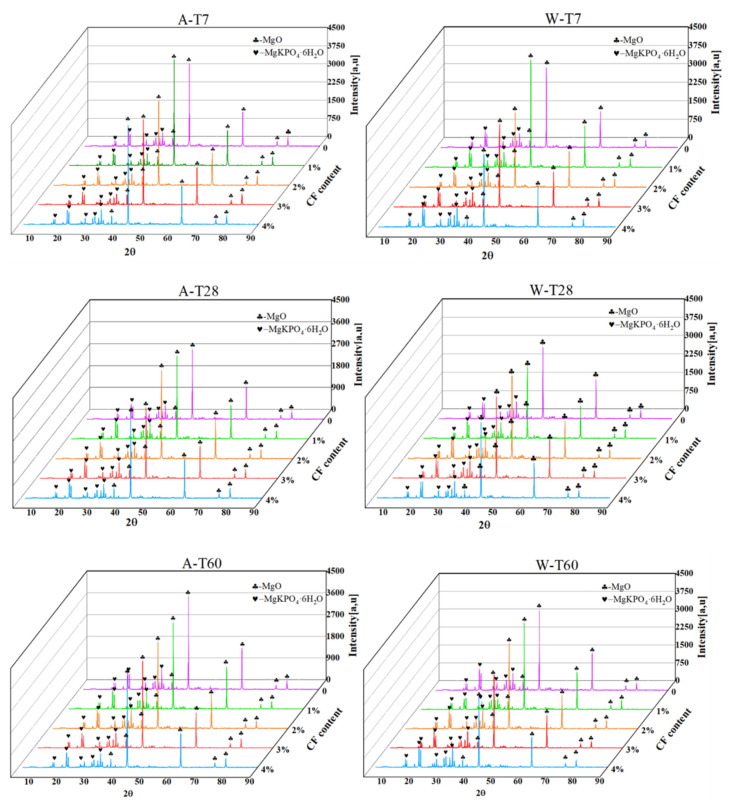
Summary of XRD spectra of CF-MPC.

**Figure 11 polymers-14-05339-f011:**
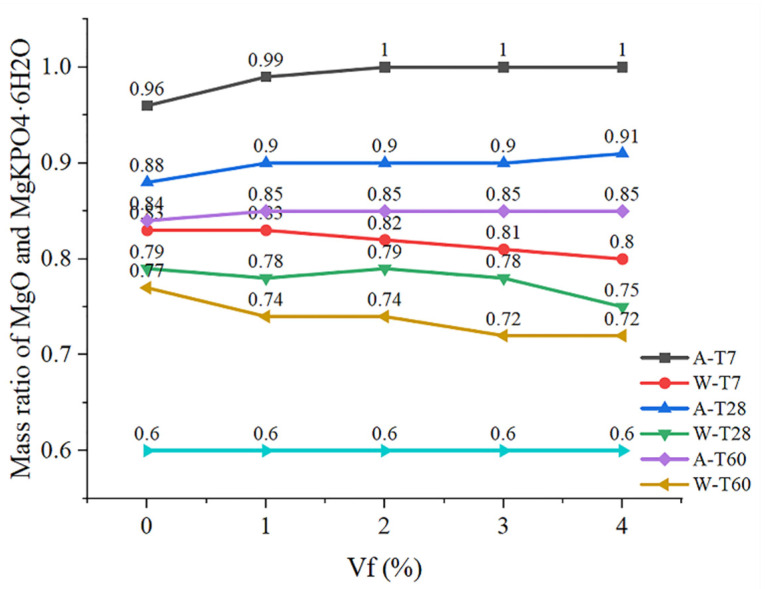
Mass ratio of MgO to MgKPO_4_·6H_2_O.

**Figure 12 polymers-14-05339-f012:**
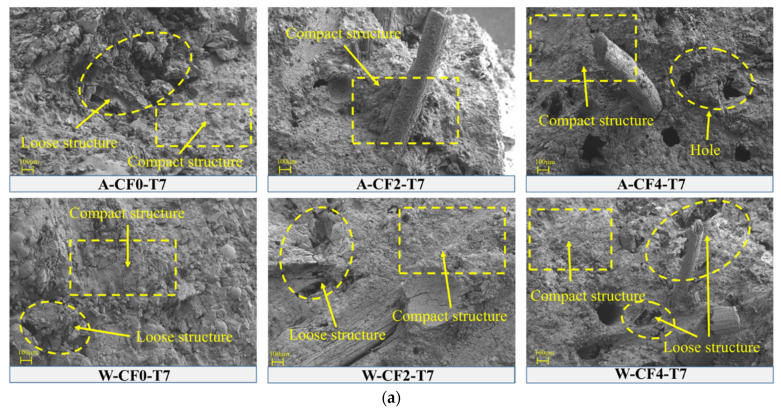
Microstructure of CF-MPC under different curing times and fiber contents. (**a**) 7 days; (**b**) 28 days; (**c**) 60 days.

**Table 1 polymers-14-05339-t001:** Chemical composition and content of magnesium oxide.

Component	MgO	CaO	SiO₂	Fe₂O₃	Al₂O₃	Other
Content (%)	96.25	1.18	1.16	1.09	0.29	3.29

**Table 2 polymers-14-05339-t002:** Chemical composition and content of fly ash.

Component	SiO₂	Al₂O₃	CaO	Fe₂O₃	K₂O	Other
Content (%)	54.94	34.86	2.63	2.52	1.76	3.29

**Table 3 polymers-14-05339-t003:** CF performance parameters.

Length (mm)	Average Diameter (μm)	Density (g/cm^3^)	Elasticity (GPa)	Tensile Strength (MPa)	Elongation at Break (%)
20	250	1.2	3.86–5.6	128–157	21.2–40.7

**Table 4 polymers-14-05339-t004:** MPC basic mix ratio parameters.

Element	MgO	KH_2_PO_4_	Borax	Fly Ash	Water
Content (g/cm^3^)	1.17	0.80	0.12	0.30	0.34

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
