# Peer review of "Effect of Water Immersion on Compressive Properties of Coir Fiber Magnesium Phosphate Cement"

_polymers, 2022, doi:10.3390/polym14245339_

Round 1
Reviewer 1 Report
that cured in air and water both have better integrity. However, fig 7a and 7b show the specimens prepared without CF addition have the highest compressive strength, cured in air and water. Please explain the reason of this results.
2. in Section 4.1 (p.10 and fig 9), the formation of MgKPO4·6H2O may results in the increase of porosity of specimen and reduce the compactness. In fig 11, the mass ratio of MgO and MgKPO4-6H2O seems not affected by the addition of VF? Why the porosity and the compactness of specimens not affect the compressive strength of the specimens.
3. From fig 7, the influence of VF and curing days seems have no obvious influence on the strength of the specimens prepared with various VF addition. Then the test results can support the purpose of this study?
4. By comparing fig 7.a and fig 7b, the strength of specimens cured in air are higher than those cured in water. The function of VF addition seems not improve the property of MPC.
Author Response
We highly appreciate the reviewers having taken the time to read our paper.Attached is our reply to your questions. In addition, we found an editing error in formula 2, and we have corrected it.

Reviewer 2 Report
The manuscript under review presents very interesting information on new solutions in the field of special binding materials, which include phosphate-magnesium cements. The physicochemical properties specific to these cements are in my opinion a very prospective area of research. Addressing the issue of the lack of resistance to bending, associated with the brittle fractures of such binders and the motivation for the application of natural coconut fibers by the authors, definitely fits the theme of sustainable development. The research results included in the paper are indicative of its high-quality. To improve the manuscript further, please consider the following comments:
1. Since the mechanical strength of the binder increases very rapidly and dynamically, it is recommended for use as a repair material and in broadly understood construction failures that require urgent intervention. I would therefore consider performing some investigations of the high-temperature resistance of the binder, complete with complex thermal analysis in combination with gas analysis. The obtained results would certainly of high interest to the mining industry, where such quick-setting binders have considerable application potential.
2. Fly ash (FA) was one of the reagents applied in the preparation of the phosphate-magnesium cement. In addition to the results of chemical composition analysis of this fly ash, the authors should also present data on its phase composition, including the fraction of its amorphous phase. The reason why I am suggesting this is that fly ash contains minerals, which under certain conditions and at a certain pH have binding properties.
3. Hydration processes in such binders tend to be highly exothermic. Will the different values of the thermal expansion coefficient of the fibers and the cement matrix lead to the formation of cracks in the material?
4. Would it possible to include a quantitative analysis of the XRD results for the hydrating slurries? Taking into account the difference in the intensity of the reflections attributed to individual phases in the diffraction patterns I suspect that hydration processes in different samples might proceed at different rates. Can the fibers reacting with the slurry affect its hydration?
5. The presence of magnesium oxide in the hydrating cement slurry may lead to the formation of expansive brucite at a later hydration stage. Is any follow-up research that would investigate this possibility planned?
Author Response
We really appreciate you taking the time to read our paper. Attached is our reply to your questions. Thank you very much for your interest in our research and your relevant opinions.
